# Microfluidic Generation of Amino-Functionalized Hydrogel Microbeads Capable of On-Bead Bioassay

**DOI:** 10.3390/mi10080527

**Published:** 2019-08-09

**Authors:** Seongsoo Kim, Sang-Myung Lee, Sung Sik Lee, Dong-Sik Shin

**Affiliations:** 1Division of Chemical and Bioengineering, Kangwon National University, Gangwon-do 24341, Korea; 2Scientific Center for Optical and Electron Microscopy, ETH Zurich, CH-8093 Zurich, Switzerland; 3Institute of Biochemistry, ETH Zurich, CH-8093 Zurich, Switzerland; 4Department of Chemical and Biological Engineering, Sookmyung Women’s University, 100 Cheongpa-ro 47-gil, Yongsan-gu, Seoul 04310, Korea

**Keywords:** polyethylene glycol, hydrogel microbead, microfluidic channel, on-bead bioassay, peptide synthesis

## Abstract

Microfluidic generation of hydrogel microbeads is a highly efficient and reproducible approach to create various functional hydrogel beads. Here, we report a method to prepare crosslinked amino-functionalized polyethylene glycol (PEG) microbeads using a microfluidic channel. The microbeads generated from a microfluidic device were evaluated by scanning electron microscopy (SEM), energy dispersive X-ray spectroscopy (EDS) and confocal laser scanning microscopy, respectively. We found that the microbeads were monodisperse and the amino groups were localized on the shell region of the microbeads. A swelling test exhibited compatibility with various solvents. A cell binding assay was successfully performed with RGD peptide-coupled amino-functionalized hydrogel microbeads. This strategy will enable the large production of the various functional microbeads, which can be used for solid phase peptide synthesis and on-bead bioassays.

## 1. Introduction

The solid-phase peptide synthesis (SPPS) method performed on solid supports offers a robust technology to produce chemically engineered peptide libraries [1,2,3]. It is particularly true that these well-defined libraries combined with combinatorial chemistry can be used to find bioactive molecules in the biochemical and pharmacological research area. Since Merrifield first demonstrated peptide synthesis by means of the SPPS method using crosslinked polystyrene supports in 1963, amino-functionalized polystyrene-based microbeads as polymer supports of the SPPS method are still widely used for peptide synthesis [4,5]. Because the polystyrene-based microbeads are not allowed to be utilized in aqueous solution due to their hydrophobicity, researchers improved hydrophilicity by grafting polyethylene glycol (PEG) into the polystyrene backbones [6,7]. Even though the amphiphilic microbeads have been developed so far, ideal solid supports are being explored. In addition, these microbeads are usually prepared by suspension polymerization, thus, their large polydispersity index (PDI) hampers the preparation of monodisperse microbeads, even after sieving [8,9,10,11,12].

For the production of monodisperse microbeads, microfluidics has been used over the past decade. Monodisperse microbeads can be synthesized by emulsifying the monomer mixture in the continuous phase to form droplets of pre-polymerized monomers followed by polymerizing the monomer in the droplet by physicochemical or photochemical methods [13,14,15]. The droplet polymerization can be easily performed by using a microfluidic chip with well-defined channel widths and flow rates of laminar fluids. This identical condition of laminar flow generates one drop at a time, resulting in the formation of a spherical particle upon breaking up the droplet [16]. Utilizing this robust method, the solid microbeads have been successfully used as attractive biomimetic scaffolds owing to their uniformity in size, structure, and composition [17,18,19]. These microbeads with PEG or naturally crosslinked polymers are highly biocompatible due to their important characteristics, which swell to adsorb and hold aqueous solution [20].

The goal of this work is to overcome the conventional supports for the SPPS method and bioassays. Since PEG hydrogels are considered ideal supporting materials due to their inert properties to various chemical or biological reagents and amphiphilic properties to aqueous and organic solvents, here we aim to produce crosslinked amino-functionalized PEG microbeads. Monodisperse microbeads are prepared in a microfluidic channel, and we evaluate physicochemical properties by various analytical methods, followed by a demonstration of the feasibility for cell binding assays.

## 2. Materials and Methods

### 2.1. Materials

Tetramethylethylenediamine (TEMED), ammonium persulfate (AP), poly(ethylene glycol) (PEG) (M_n_ = 200), poly(ethylene glycol) methyl ether acrylate (PEGA) (M_n_ = 480), poly(ethylene glycol) diacrylate (PEGDA) (M_n_ = 700), 2-aminoethyl methacrylate hydrochloride, fluorescein isothiocyanate (FITC), and bromophenol blue (BPB) were purchased from Sigma-Aldrich (St. Louis, MO, USA). Dichloromethane (DCM), acetonitrile (ACN), methanol (MeOH), *N*,*N*′-dimethylformamide (DMF), acetic anhydride, and *N*,*N*′-diisopropylethylamine (DIPEA) were purchased from Daejung Chemicals & Metals Co. (Gyeonggi-do, Korea). HFE 7500 (Novec, Singapore) and krytox (modified) fluorosurfactant were purchased from RAN biotechnologies (Beverly, MA, USA).

We utilized microfluidic chips from Darwin microfluidics (Cell Barcoding Chip-PDMS, Paris, France: Channel depth = 80 µm, nozzle width = 100 µm, main channel width = 140 µm). The prepolymer solution and oil solution in 1-mL syringes, respectively, were injected into chip inlets using syringe pumps (NE 1600, New Era Pump Systems, Farmingdale, NY, USA) through PTFE tubes (TUBF22-10, MISUMI, Tokyo, Japan).

Peptide (Gly-Arg-Gly-Asp-Ser-Cys; GRGDSC, 80% HPLC purity) was synthesized by the conventional SPPS method. The maleimide-*N*-hydroxysuccinimide ester bifunctional linker (MAL-dPEG_6_-NHS ester: MAL-NHS linker) was purchased from Quanta Biodesign (Powell, OH, USA). NIH 3T3 mouse fibroblasts were purchased from Korea Cell Line Bank (Seoul, Korea) and cultured in 10% (*v*/*v*) fetal bovine serum (FBS), 100 U/mL penicillin and 100 µg/mL streptomycin in RPMI 1640 media (Biowest, Nuaillé, France) at 37 °C in a humidified 5% CO_2_ atmosphere. Trypsin-EDTA in PBS was purchased from Biowest (Nuaillé, France). Calcein AM was purchased from Invitrogen (Carlsbad, CA, USA). Confocal laser scanning microscopic images were obtained by using Carl Zeiss LSM-700, Axio Observer (Oberkochen, Germany).

### 2.2. Generation of PEG-Amine Hydrogel Microbeads

PEG hydrogel microbeads were synthesized through microfluidic encapsulation. The aqueous phase of TEMED (60 mg) was dissolved in deionized (DI) water (370 μL). It was then mixed with PEGDA (330 μL) and PEGA (300 μL) as a prepolymer solution for the first inlet. The second inlet aqueous solution was composed of PEGA (100 μL), AP (50 mg), and DI water (900 μL). The third inlet aqueous solution contained PEG (100 μL) and 2-aminoethyl methacrylate hydrochloride (35 mg) in DI water (900 μL). Krytox (0.5%, modified) surfactant was dissolved in the oil phase (fluorocarbon oil HFE 7500). The solidified droplets by AP/TEMED-catalyzed polymerization were collected in a glass vial at the outlet. The flow rates of the aqueous phase (50 μL/h) and oil phase (250 μL/h) were controlled by using individual syringe pumps. The PEG hydrogel microbeads were separated from droplets by adding 20% perfluorooctane in HFE oil and sequentially washed with 1% (*v*/*v*) Span 80 in hexane, 0.1% (*v*/*v*) triton X-100 in hexane, and DI water three times, respectively. The microbeads were analyzed by scanning electron microscopy (SEM), energy dispersive X-ray spectroscopy (EDS), and confocal laser scanning microscopy, respectively.

### 2.3. Coupling of FITC onto PEG-Amine Hydrogel Microbeads

A reaction tube containing PEG-amine beads (10 mg) was mixed in the dark with a solution of FITC (2 equivalent) and DIPEA (4 equivalent) in DMF (3 mL) and shaken for 18 h. The PEG bead was washed with DMF, DCM, and MeOH three times, respectively, and dried in vacuo. Fluorescence images of the microbeads were obtained by using a confocal laser scanning microscope (CLSM).

### 2.4. RGD Peptide Immobilization on Peg Hydrogel Microbeads

To a solution of 3.4 μM MAL-NHS linker and 6.8 μM DIPEA in DMF (1 mL), PEG hydrogel microbeads (13 mg) were added and incubated for 1 h at 37 °C. After three alternate washes with fresh DMF and MeOH, the hydrogel microbeads were re-incubated in a solution of 1.6 μM GRGDSC peptide and 0.6 μM DIPEA in DMF (1 mL) for 1 h at 37 °C. Finally, the peptide immobilized hydrogel microbeads were washed with DMF and methanol, then stored at a vacuum desiccator.

### 2.5. Cell Binding Assay

The NIH 3T3 cells (300,000 cells/mL) were trypsinized and incubated with the peptide- immobilized hydrogel microbeads (1 mg) in culture media within a 96-well plate for 2 d. The microbead-bound cells were stained with Calcein AM solution (1 μM) in PBS for 40 min. The cells on beads were stained with Calcein AM were observed under an optical microscope and a CLSM for peptide-induced cell behavior on bead surfaces.

## 3. Results and Discussion

A microfluidic device is designed to offer three aqueous channels and a single oil channel (Figure 1). As a control experiment, poly(ethylene glycol) methyl ether acrylate (PEGA) crosslinked with poly(ethylene glycol) diacrylate (PEGDA) was encapsulated in droplets (Figure 1a). To functionalize hydrogel microbeads with amino groups, an aqueous solution of 2-aminoethyl methacrylate was added into the third inlet channel (Figure 1b). The addition of the third inlet channel led to immediate earlier AP/TEMED-catalyzed polymerization of monomers (PEGA and PEGDA) at the core region, followed by surface-localized amino group functionalization preventing 2-aminoethyl methacrylate polymerization in the core region. More specifically, since a solution of PEGA and PEGDA coexisted with TEMED, the radical initiation preceded at the core region. The injection of 2-aminoethyl methacrylate without TEMED would induce a delayed polymerization because reactive radicals were derived from the core region, which led to the generation of core-shell type microbeads. The size of the droplets generated from a cross-flow generator (Figure 1(b1)) was determined by the ratio of flow rates and the dimension between aqueous and oil channels (Figure 1(b1,b2)). The PEG-amine hydrogel microbeads formed in a spherical shape at the outlet of the channels (Figure 1(b3)).

After collecting droplets (Figure 2a) and the emulsified droplets were destabilized, we observed that the microbeads maintained the spherical structure (Figure 2b) and were highly monodisperse (standard deviation (SD) of microbead size: ca. 7%) (Figure 2e). After the PEG-amine microbeads were swollen in water, their diameter expanded from 79.5 to 104.1 µm by 31% (Figure 2f). As seen from scanning electron microscope (SEM) images, the surface of an amino-functionalized bead was bumpy (Figure 2c), while the microbead without amino groups presented a glazed surface (Figure 2d).

From a bromophenol blue (BPB) staining result, the blue color exhibited amino groups stained by BPB (Figure 3a). The PEG-amine microbeads have a discrete interface between the surface and the core regions, and the thickness of the amino-functionalized shell was measured as ca. 8 µm. The amino groups were coupled with FITC, and the CLSM fluorescence images at different z-levels of the FITC coupled microbeads strongly supported that the presence of amino groups were only at the shell region (Figure 3b) [21]. Furthermore, a three-dimensional reconstructed z-stack image visualized core-shell typed amino microbeads (Figure 3c). As another approach of local elemental analysis of the microbeads, energy dispersive X-ray spectroscopy (EDS) was employed to evaluate the presence of amino groups (Figure 4). The elemental concentration of nitrogen from the PEG-amine microbead surface was measured at ca. 12%, which was 6-fold larger than the PEG microbead surface (Figure 4).

To evaluate solvent compatibility of our PEG-amine microbeads, the swelling volume was measured in various solvents (Table 1). The PEG-amine microbeads exhibited a similar swelling volume per weight of the microbeads in polar and nonpolar solvents, such as toluene (1.8 mL/g), DMF (2.5 mL/g), DCM (3.3 mL/g), ACN (3.3 mL/g), methanol (2.4 mL/g), and water (2.2 mL/g). We found that the non-functionalized PEG microbeads were more compatible with various solvents due to the increase in amphiphilicity [22]. Compared with commercially available PEG-based microbeads, such as PEGA resin [23,24], TentaGel XV resin [25,26], and ChemMatrix resin [27,28], PEG-amine hydrogel microbeads did not swell relatively with various solvents because of the high degree of crosslink [4,5]. However, our PEG-amine microbeads did not need a high swelling ratio because most of the active functional groups were exposed at the shell region [22,29]. Since amino groups of PEG-amine microbeads are located at the shell of the beads, the accessibility of the reagents are already ensured to react with the amino groups without diffusion into the core region [29,30,31].

To examine the cellular behavior on amino-functionalized PEG hydrogel microbead, NIH 3T3 cells were seeded on an RGD sequence containing peptide (GRGDSC) immobilized PEG-amine microbeads. After 72 h, cell morphology is confirmed as a spherical shape after Calcein AM staining (Figure 5d–f). NIH 3T3 cells tended to crouch on the PEG surface rather than develop lamellipodia, because a high content of PEG prevented spreading on the surface of the beads, even though there was a presence of RGD-dependent extracellular matrix (ECM) integrin interaction in the PEG domain. However, the cells still attached onto the RGD-immobilized microbeads. In contrast, there was no cell attachment on the none-RGD functionalized PEG beads at 72 h after cell seeding (Figure 5a–c). We could observe that the microbeads immobilized with a cell binding peptide enabled cell binding on microbeads without nonspecific adsorption. These results showed that the binding assays on the PEG-amine microbeads are feasible [29,32,33].

## 4. Conclusions

In this study, amino-functionalized hydrogel microbeads were prepared in a microfluidic channel. The sequential aqueous phase injections created amino-functionalized hydrogel microbeads. Microscopic images and elemental analysis by EDS revealed that the microbeads were monodisperse and the amino groups existed on the surface of the microbeads. While NIH 3T3 cells did not attach on non-peptide microbeads, cells attached onto RGD-coupled amino-functionalized hydrogel microbeads. In the future, these functionalized PEG microbeads will be utilized to capture specific cells in whole blood to minimize the interaction with various nonspecific hematocytes and proteins by strong non-fouling properties of PEG. We envision that the microbeads can be used for SPPS and bioassays with high efficiency. In addition, this microfluidic approach enables the large- scale production of functionalized microbeads with a parallel multichannel synthesis system.

## Figures and Tables

**Figure 1 micromachines-10-00527-f001:**
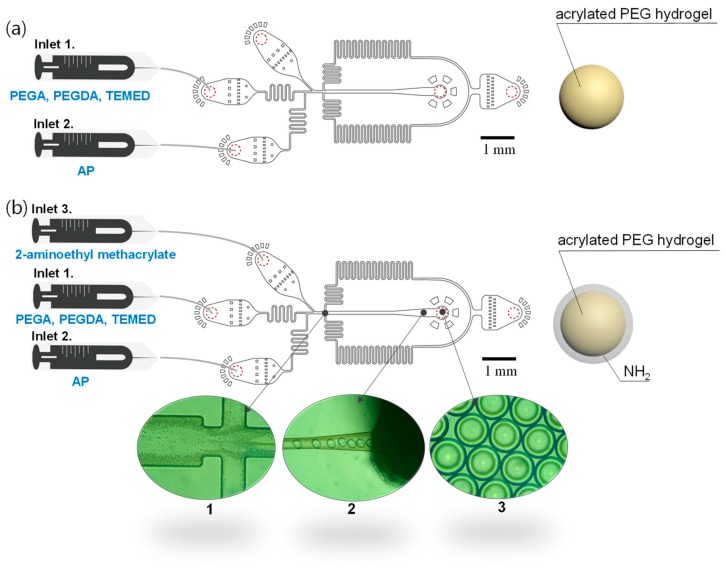
Design of microfluidic channels for the generation of PEG-amine microbeads: (**a**) Non-functionalized PEG microbeads using two inlet channels and (**b**) amino-functionalized PEG-amine microbeads using three inlet channels. (**b1**) Flow focus junction, (**b2**) hydrogel precursor droplets flowing in channel, and (**b3**) emulsified droplets at the outlet.

**Figure 2 micromachines-10-00527-f002:**
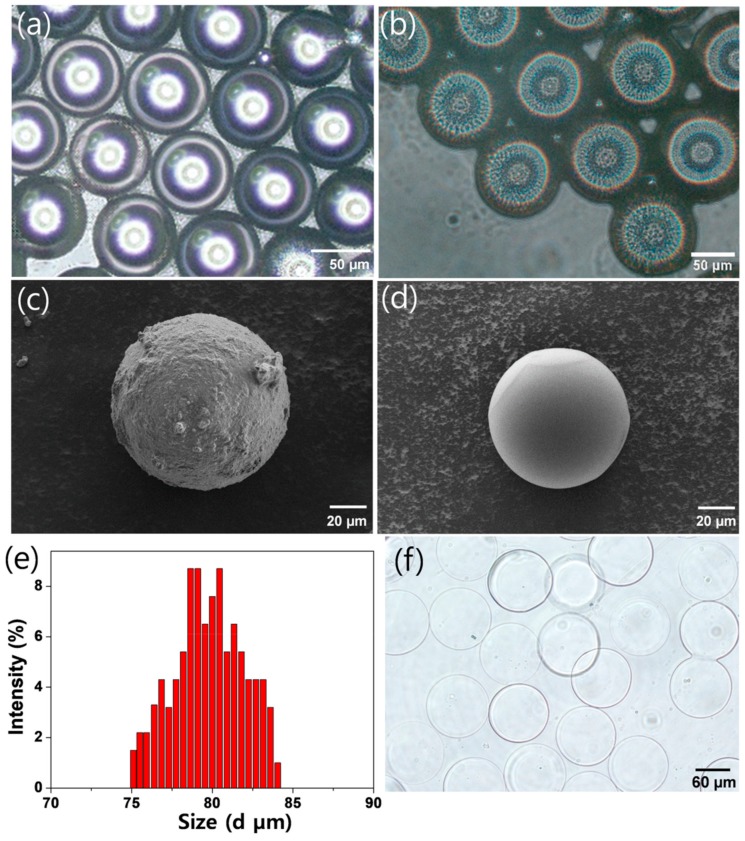
Microscopic images of (**a**) emulsified hydrogel precursor droplets and (**b**) PEG-amine microbeads after destabilization. Scanning electron microscopic (SEM) images of (**c**) PEG-amine microbeads and (**d**) non-functionalized PEG microbeads. (**e**) Size distribution of PEG-amine microbeads (100 ea) and (**f**) microscopic image of PEG-amine microbeads swollen in water.

**Figure 3 micromachines-10-00527-f003:**
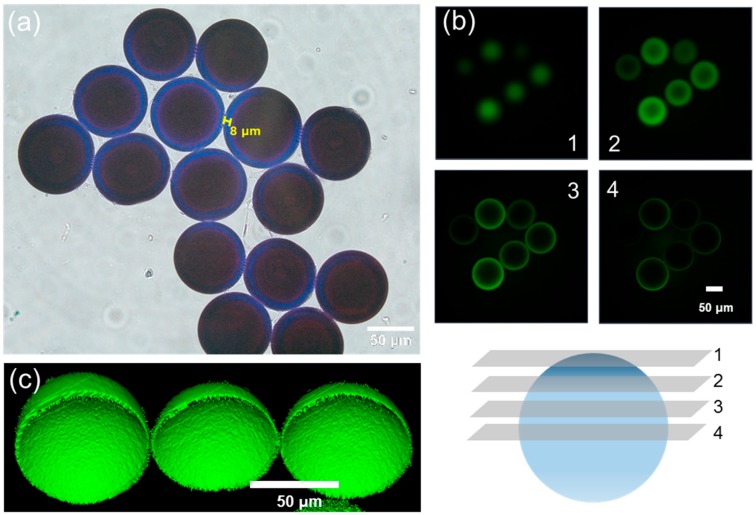
(**a**) Microscopic image of PEG-amine microbeads after bromophenol blue (BPB) staining. (**b**) Confocal laser scanning microscopic (CLSM) images of fluorescein isothiocyanate (FITC)-labeled PEG-amine microbeads at different z-levels and (**c**) three-dimensional reconstructed z-stack image.

**Figure 4 micromachines-10-00527-f004:**
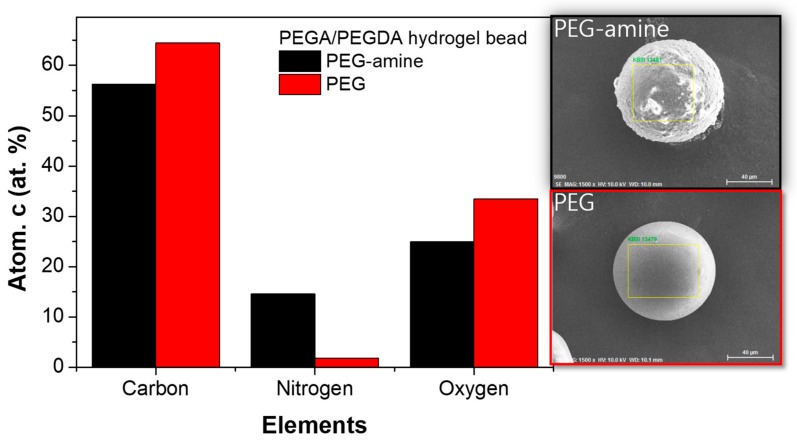
Energy dispersive X-ray spectroscopy (EDS) analysis of PEG-amine microbeads (black) and non-functionalized PEG microbeads (red).

**Figure 5 micromachines-10-00527-f005:**
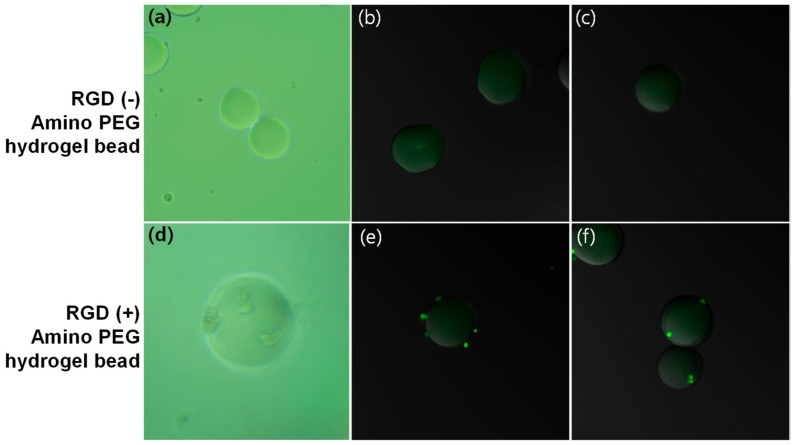
(**a**,**d**) Microscopic images and (**b**,**c**,**e**,**f**) confocal laser scanning microscope images of cell- treated hydrogel microbeads: NIH 3T3 cells were not attached on the PEG-amine beads (top) while cells were attached on GRGDSC peptide-immobilized PEG-amine microbeads (bottom).

**Table 1 micromachines-10-00527-t001:** Swelling volume of PEG microbeads with or without amine functionalization in the various solvents.

Swelling Volume (mL/g)	Toluene	ACN	DCM	DMF	MeOH	Water
PEG-amine microbead	1.8	3.3	3.3	2.5	2.4	2.2
Non-functionalized PEG microbead	3.4	4.2	4.6	4.0	3.1	2.5

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
