# Peer review of "Microfluidic Generation of Amino-Functionalized Hydrogel Microbeads Capable of On-Bead Bioassay"

_micromachines, 2019, doi:10.3390/mi10080527_

Round 1

Reviewer 1 Report

l. 63: Singapore

The central part of Fig. 1 is nearly black and difficult to read. Perhaps a different format is needed.

Fig. 3b and 3c are very dark and of poor contrast. This should be improved.

l. 188 "did not attached" attach?

In Fig. 4, the x-axis should read elements instead of atoms.

Where does the irregularity of the PEG-amine beads come from?

"However, our PEG-amine microbeads do not need high swelling ratio because most of active functional groups are exposed at the shell region." I do not understand this argument. Nearly all peptide supports need to be swollen to achieve a good capacity, good steric accessibility and good synthesis yields.

In general, the paper is fine and does not need too much revisions. However, I have some more fundamental questions:

If the main application should be SPPS, a core of pure PEG would be inactive. The peptide capacity would be low. It would be quite important to determine the amino capacity of the beads in relation to other beads. At which group the peptide linker would be attached? Perhaps a model peptide synthesis should be performed.

The authors only show one size of the beads. Would the technique be flexible enough, to produce beads of another diameter?

Are the beads physically stable or soft?

Reviewer 2 Report

In the article “Microfluidic generation of amino-functionalized hydrogel microbeads capable of on-bead bioassay” Kim et al. describe a microfluidic-based method for the generation of micro-particles that potentially could be used for solid-phase peptide synthesis or bioassays.

The Reviewer believes that the manuscript could be suitable for publication in Micromachines after the following major revisions:

1 – it is not clear to the Reviewer the actual method used for bead production. The Authors mentioned in paragraph 2.2 that “The prepolymer droplets were collected in a glass vial at outlet followed by AP/TEMED-catalyzed polymerization”. That means that the crosslinking of the particles happened outside the chip. If so, the injected solutions have enough time to mix.

Thus, the Reviewer has major doubts about the presented core-shell structure of the particles. In view of the aforementioned Reviewer’s consideration, how a core-shell structure is achieved in the end? what is the actual mechanism? The Authors should address this point in details providing a clear explanation of the microparticle production mechanism.

Additionally, the three pre-polymer streams do not have a core-shell compartmentalization in the microfluidic chip – thus how would it be possible to obtain core-shell particles?

2 – Figure 3 – In the Reviewer’s opinion, the presented images are not enough to demonstrate their thesis – i.e. the core-shell structure of the particles. Did the Authors consider any problem related with the diffusion of the dyes? 

3 – Figure 4 – Studying the surface of the microbeads is not enough to demonstrate the core-shell structure of the beads. The Authors should repeat quantitative EDS experiment along a microbead cross-section showing precisely the “borders” between amine-reach shell and core region.

4 – Figure 5 – The presented results are not really satisfying. The Authors should study not only cell attachment but also proliferation, with eventual full colonization of the microbead surface.

Round 2

Reviewer 1 Report

"In addition, for the industrial peptide synthesis, excessive swelling volume of the polymer beads lead to spend a lot of energy and financial resources to transport and recycle it. If the bead is low bed-volume to substitution ratio, reagents and solvents are minimized, thus, it is efficient for large scale batch synthesis system."

I do not agree with this statement. Since there is no experimental proof given for this, I would delete this section.

Ref. 32 seems to be not formatted and numbered correctly.

I am still not completely convinced of the concept, since practical applications are still missing and the claimed advantages are not shown experimentally. In a next paper, a direct comparison with commercial resins, such as Tentagel and others, should be shown. However, this paper might be published as a "work in progress".

Reviewer 2 Report

The Authors have partially addressed my remarks in the current manuscript version. However, few points must be still clarified in order to improve the quality and impact of the article.

1 - Since the Authors did not provide any new data about EDS experiment to clarify the core-shell structure of the beads - the Reviewers expects a confocal Z-stack in which it could be clearly seen the core-shell structure of the presented beads.

2 - Could the Authors discuss further in the article how core-shell particles are obtain using a co-planar fluid streams? Is it a matter of cross-linking kinetics of the two solutions? This is really unclear in the Reviewer's opinion.

After addressing these remarks, the article should be ready for acceptance.
